# Actinomycetes Enrich Soil Rhizosphere and Improve Seed Quality as well as Productivity of Legumes by Boosting Nitrogen Availability and Metabolism

**DOI:** 10.3390/biom10121675

**Published:** 2020-12-15

**Authors:** Hamada AbdElgawad, Walid Abuelsoud, Mahmoud M. Y. Madany, Samy Selim, Gaurav Zinta, Ahmed S. M. Mousa, Wael N. Hozzein

**Affiliations:** 1Botany and Microbiology Department, Faculty of Science, Beni-Suef University, Beni-Suef 62511, Egypt; hamada.abdelgawad@science.bsu.edu.eg (H.A.); ahmed.mousa@science.bsu.edu.eg (A.S.M.M.); 2Department of Botany and Microbiology, Faculty of Science, Cairo University, Giza 12613, Egypt; walidabc@sci.cu.edu.eg (W.A.); madany@cu.edu.eg (M.M.Y.M.); 3Biology Department, College of Science, Taibah University, Al-Madinah Al-Munawarah 41411, Saudi Arabia; 4Department of Clinical Laboratory Sciences, College of Applied Medical Sciences, Jouf University, Sakaka P.O. 2014, Saudi Arabia; sabdulsalam@ji.edu.sa; 5Shanghai Center for Plant Stress Biology, Center of Excellence in Molecular Plant Sciences, Chinese Academy of Sciences, Shanghai 200032, China; gaurav@sibs.ac.cn; 6Bioproducts Research Chair, Zoology Department, College of Science, King Saud University, Riyadh 11451, Saudi Arabia

**Keywords:** actinomycetes, legumes, photosynthesis, crop yield, nitrogen metabolism, biofertilizer

## Abstract

The use of actinomycetes for improving soil fertility and plant production is an attractive strategy for developing sustainable agricultural systems due to their effectiveness, eco-friendliness, and low production cost. Out of 17 species isolated from the soil rhizosphere of legume crops, 4 bioactive isolates were selected and their impact on 5 legumes: soybean, kidney bean, chickpea, lentil, and pea were evaluated. According to the morphological and molecular identification, these isolates belong to the genus Streptomyces. Here, we showed that these isolates increased soil nutrients and organic matter content and improved soil microbial populations. At the plant level, soil enrichment with actinomycetes increased photosynthetic reactions and eventually increased legume yield. Actinomycetes also increased nitrogen availability in soil and legume tissue and seeds, which induced the activity of key nitrogen metabolizing enzymes, e.g., glutamine synthetase, glutamate synthase, and nitrate reductase. In addition to increased nitrogen-containing amino acids levels, we also report high sugar, organic acids, and fatty acids as well as antioxidant phenolics, mineral, and vitamins levels in actinomycete treated legume seeds, which in turn improved their seed quality. Overall, this study shed the light on the impact of actinomycetes on enhancing the quality and productivity of legume crops by boosting the bioactive primary and secondary metabolites. Moreover, our findings emphasize the positive role of actinomycetes in improving the soil by enriching its microbial population. Therefore, our data reinforce the usage of actinomycetes as biofertilizers to provide sustainable food production and achieve biosafety.

## 1. Introduction

Conventional farming has contributed immensely to increase crop yield and productivity to feed the world’s growing population. However, the excessive usage of fertilizers and agrochemicals has caused environmental and health concerns. Organic farming is emerging as an alternative to conventional agricultural practices. The new trend embraces the use of natural compounds and/or biofertilizers as a replacement of synthetic chemical inputs to improve plant growth and productivity, and increase stress resistance [1,2,3,4,5]. Moreover, natural compounds and biofertilizers can also biosynthesize nanoparticles, which could reduce the use of synthetic fertilizers [6].

Actinomycetes are a group of bacteria with high guanine–cytosine content in their genomic DNA and some of them are filamentous with true mycelia. They are widely spread in different habitats including soil environments, where they are involved in dead organic matter decomposition and nitrogen fixation (around 15% of nitrogen fixation is done by actinomycetes), and in phosphates solubilization. Actinomycetes are well known to produce a variety of antibiotics, biocontrol agents, and plant growth-promoting chemicals. [2,4,5,7,8,9]. Moreover, they were reported to promote plant growth by indirect and/or direct modes. The indirect mode involves growth suppression of pathogenic organisms by actinomycetes [10]. For instance, actinomycetes suppress the growth of fungal pathogens *Pythium ultimum* and *Erwinia carotovora*, which cause damping-off and post-harvest rot disease, respectively [11]. Actinomycetes achieve antagonism by excreting 1,3-β-glucanase and cell wall hydrolyzing enzymes [12]. On the other hand, actinomycetes directly impact plant growth by excreting plant growth-promoting compounds, mineral nutrients, or enhancing the growth of beneficial microorganisms. *Streptomyces* from chickpea rhizosphere produce siderophore, indole acetic acid (IAA), lytic and cellulase enzymes, and are linked with the increase in nodule number and growth of chickpea [5,13]. Actinomycetes treatment is also known to improve growth of several plants through the production of gibberellic acid and IAA [14]. Similarly, isolates of *Streptomyces*, the largest genus of actinomycetes, were reported to promote the growth of both tomato and maize, due to their ability to produce siderophores [8,15]. Moreover, other genera of actinomycetes like *Actinomadura, Micromonospora*, *Streptosporangium,* and *Nocardia* can ameliorate the mineralization of organic phosphates through the secretion of alkaline or acidic phosphatase enzymes [2]. Concomitantly, *Micromonospora endolithica* promotes the growth of *Phaseolus vulgaris* through the solubilization of organic phosphates as well as increasing K, Mg, S, Fe, and N availability [16]. Wheat grains coated with *Micromonospora aurantiaca* and *Streptomyces griseus* mycelia showed improvement in wheat plant growth, mainly due to increased mineral contents (e.g., N and P) in wheat plant. This was a result of phosphate solubilizing activities of applied actinomycetes; the mycelial coating also protected wheat grains from damping-off disease caused by *Pythium ultimum* [11].

Actinomycetes have been shown to be involved in nitrogen fixation in various legumes and non-legumes without forming nodules [2,17]. Thus, actinomycetes have a huge influence on nitrogen availability and flux in the air–soil–plant system. Moreover, actinomycetes can affect plant photosynthesis, carbon, and nitrogen metabolism. For instance, *Streptomyces coelicolor* HHFA2 enhances the photosynthetic pigments content and foliar growth parameters of onion in pots and field studies [4]. In the study of Weston et al. [18], transcriptome analysis and metabolic profiling of shoots of *Arabidopsis* co-cultured with two strains of *Pseudomonas fluorescens* were enriched with transcripts and metabolites of secondary metabolism, photosynthesis as well as amino acids metabolism. In *Frankia*-infected *Datisca glomerata* plants, cytosolic glutamine synthetase was highly abundant, thus improving *D. glomerate* growth [19]. Kurth et al. [20] showed that a *Streptomyces* inoculum suppressed the powdery mildew infection of oak and alleviated the depletion of photosynthesis-related transcripts in response to infection, thus provided protection for oak plants upon pathogen challenge.

The above-mentioned examples clearly provide some links between actinomycetes and the increase in plant growth and productivity. Besides the scarcity in details about the impact of actinomycetes on soil properties as well as plant metabolic processes, the implications of actinomycetes upon seed nutritional profiles are not clear. Moreover, there is an urgent need for eco-friendly alternatives to agrochemicals to ensure biosafety. Therefore, we conducted this study to investigate the potentiality of the most bioactive isolates of actinomycetes on growth as well as physiological and metabolic behavior of five major leguminous crops.

## 2. Materials and Methods

### 2.1. Isolation of Actinomycetes

Soil samples, collected from the local legume fields in Jouf region (Sakaka, Saudi Arabia), were used for isolation of the actinomycetes. To extract actinomycetes, 1 g of soil was mixed with 10 mL of medium and placed in a stirring water bath (SW23 Shaking Water Bath, JULABO, Seelbach, Germany) for 20 min. The suspension was centrifuged for 5 min at 4000× *g* to remove coarse soil particles. The supernatant, containing actinobacteria was diluted and placed on agar media [21]. glycerol–yeast extract agar (glycerol 5 mL, yeast extract 2 g, K_2_HPO_4_ 1 g, agar 15 g, distilled water 1000 mL) containing nystatin (50 μg/mL) was used as the isolation medium [22]. The isolation plates were incubated at 28 °C for 14 days. Actinomycetes colonies were purified and incubated at 28 °C for 7 days. The pure cultures of actinomycetes were maintained in 20% glycerol suspensions at −20 °C.

### 2.2. Morphological and Biochemical Characterization of the Isolated Actinomycetes

Characteristics of the isolates were recorded on ISP 2 medium [23]. Morphological observations of the spore-bearing hyphae, spore chain, and spore morphology were made by using the coverslip technique of Kawato and Shinobu [24]. The cover slip was stabbed onto the agar (at 45° angle) and incubated at 28 °C for 14 days. Cover slips were then taken out from the agar and put onto the prepared slides and the staining Crystal Violet dye was used. Spore arrangement and spore surface ornamentation were observed by examining gold coated dehydrated preparations using a JEOL (JSM-6060 LV, Tokyo, Japan) scanning electron microscope. Various biochemical tests were performed according to Oskay et al. [25] to characterize the isolates.

### 2.3. Molecular Identification of the Biologically Active Actinomycete Strains

For the molecular identification, DNA was isolated from the selected 4 strains using the DNeasy UltraClean Microbial Kit by Qiagen (Hilden, Germany) according to the manufacturer’s instructions. Amplification of 16S rRNA by PCR, purification of PCR products, and sequencing were done according to Hozzein and Goodfellow [26]. The 16S rRNA sequences were compared with available sequences in the GenBank database using the BLAST tool [27]. Multiple alignments were carried out (MEGA version 7 [28]) and phylogenetic trees were generated [29]. Tree topologies were evaluated by bootstrap analysis [30] based on 1000 resamples.

### 2.4. Experimental Setup, Plant Materials and Growth Conditions

Twenty milliliters of the actinomycete culture in the log phase (OD_600_ = 1) containing the selected isolates (2, 8, 12, or 15) were mixed with 0.5 kg of dried soil potting mix (1/25; *w*/*v*) (Tref EGO substrates, Moerdijk, Netherland) in 25 × 25 cm pots. The inoculated moist soil was kept in dark at 30 °C for 1 day. Non-treated soils served as control. Soybean (*Glycine max*) var. Giza 22, kidney bean (*Pisum sativum* L.) var. Giza 3, chickpea (*Cicer arietinm L.*) var. Giza 531*,* chickpea, lentil (*Lens culinaris*) var. Sinai 1, and pea (*Pisum sativum* L.) var. Nebraska seeds were surface sterilized for 1 min in 70% ethanol and rinsed well (3 times) with sterile desilted water and then sown in treated and non-treated soils. Pots were transferred to a controlled-growth room (21/18 °C air temperature, 150 μmol PAR m^−2^ s^−1^, 16/8 h day/night photoperiod, and 60% humidity). At the end of the experiment, treated and control soils were collected for physical and biochemical analyses. Mature dried seeds were harvested from soybean, pea, chickpea, kidney bean, and lentil plants after 100, 95, 100, 105, and 110 days of sowing, respectively, and used for metabolic profiling.

### 2.5. Soil Analysis

Chemical properties of the soil (pH and electrical conductivity) were measured in diluted aqueous soil extract (1:5 *w*/*v*) using a pH meter (AD 3000) and a conductivity meter (Jenway 3305), respectively. Following Zhang et al. [31]), the phenolic content was estimated spectrophotometrically (Shimadzu UV-Vis 1601 PC, Kyoto, Japan). Twenty grams of soil were stirred vigorously with bi-distilled water followed by filtration and the filtrate was used for estimation of phenolic content. The content of mineral nutrients in soil rhizosphere were determined by shaking the excavated root very gently to separate from the bulk soil; then, the soil attached to the fine roots (2 mm thick layer) was obtained by brushing. About 20 g of soil rhizosphere were extracted in aqueous HNO_3_ solution (80%) and were detected by using ICP-MS (Finnigan Element XR, Scientific, Beremen, Germany) [32]. A total of 20 g of soil was mixed in a flask with 100 mL of distilled water and shaken for 12 h. Then, the soil extract was filtered and used to determine organic acid and phenolic compounds [33]. The percentage of CaCO_3_ was estimated according to Brown et al. [34]. The contents of carbon (C) and nitrogen (N) were measured by CN element analyzer (NC-2100, Carlo Erba Instruments, Milan, Italy).

### 2.6. Microbial Counts

Ten grams of each soil sample was added to 95 mL of 0.1% (*w*/*v*) solution of sodium pyrophosphate. After homogenization for 30 min, this solution was decimally diluted (10^−1^ to 10^−7^) and aliquots of the resulting solutions plated on appropriate culture media included tryptone soy agar for total microbial count, MacConkey agar for coliform counts, glycerol–yeast extract agar for actinomycetes counts, and Czapek Dox agar for fungi counts [22]. After incubation at 25 or 30 °C for up to10 days, the colony-forming units (CFU) were counted.

### 2.7. Photosynthesis Rate and Pigment Analysis

The rates of photosynthesis (light saturated) and respiration (expressed as µmol CO_2_ m^−2^ s ^−1^) of legumes leaves were measured by LI-COR LI-6400 [35]. Chlorophyll concentration in fresh leaf samples was measured after extraction in 80% acetone [36].

### 2.8. Metabolite Profiling

#### 2.8.1. Metabolite Profiling of Isolated Strains

Total phenolic content and flavonoids were extracted in 80% ethanol (*v*/*v*) using a MagNALyser (Roche, Vilvoorde, Belgium). Phenolics and flavonoids levels were assayed following Folin–Ciocalteu and aluminum chloride calorimetric assays, respectively [37]. Flavonoid profile was quantified using a Shimadzu HPLC system (SCL-10A vp, Shimadzu Corporation, Kyoto, Japan) and detected by UV detector, by using the mobile phase (water–formic acid (90:10, *v*/*v*) and acetonitrile/water/formic acid (85:10:5, *v*/*v*/*v*)) [22].

#### 2.8.2. Metabolite Profiling of Isolated Treated Legumes

At primary metabolic levels, 100 mg of dried powdered seed was homogenized in 80% ethanol by using a MagNALyser. Extracted total sugars in the solvent extract was measured according to Nelson’s method and Folin–Ciocalteu assay) [38]. Organic acids were extracted in extraction solvent (0.3% (*w*/*v*) of butylated hydroxyanisole in 0.1% phosphoric acid) using a MagNALyser. Then, different individuals of organic acids were separated on SUPELCOGEL C-610H column and phosphoric acid (0.1% *v*/*v*); a flow rate of 0.45 mL/min was used, and then organic acids were detected by a UV detection system [36].

Amino acids in 100 mg of treated and non-treated seeds were extracted in 80% aqueous ethanol using [37]. After centrifugation at 12,000 rpm for 30 min, the extracted amino acids were separated and quantified using Waters Acquity UPLC-tqd system (Milford, Worcester County, MA, USA) equipped with a BEH amide 2.1 × 50 column. Fatty acids were extracted from dried powdered grains (100 mg) in 50% aqueous methanol at 25 °C. GC/MS analysis was used for identification (Hewlett Packard 6890, MSD 5975 mass spectrometer (Hewlett Packard, Palo Alto, CA, USA)) with an HP-5 MS column (30 m × 0.25 mm × 0.25 mm) [37]. Fatty acids were identified using NIST 05 database and Golm Metabolome Database (http://gmd.mpimp-golm.mpg.de). Phenolics and flavonoids were extracted in acetone–water solution (4:1 *v*/*v*) at room temperature. After centrifugation, phenolic and flavonoid compounds were quantified using a Shimadzu HPLC system (SCL-10A vp, Shimadzu Corporation, Kyoto, Japan) and detected by UV detector, by using the mobile phase (water–formic acid (90:10, *v*/*v*) and acetonitrile/water/formic acid (85:10:5, *v*/*v*/*v*)) [37].

At vitamin level, tocopherols were extracted grains using hexane solvent [22]. After centrifugation, extracted tocopherols were separated and quantified by HPLC (Shimadzu, Hertogenbosch, The Netherlands) using normal phase conditions (Particil Pac 5 µm column material, length 250 mm, i.d. 4.6 mm). Phylloquinone was extracted in extraction buffer containing 90% methanol and 10% dichloromethane and 5mL of a methanolic solution containing ZnCl_2_ (1.37 g), sodium acetate (0.41 g), and acetic acid (0.30 g) per liter of the mobile phase. It was detected by HPLC using a fluorescence detector (excitation, 243 nm; emission, 430 nm). The β-cryptoxanthin contents were extracted in acetone and analyzed by a reversed phase HPLC conducted with a diode array detector [39]. Thiamine was extracted in 0.1 N HCl after incubation of 30 min in a water bath. After centrifugation, thiamine was determined on a reverse-phase (C18) column using a methanol:water mobile phase and fluorescence as a detector.

### 2.9. Minerals Content Determination

About 0.1 g of oven-dried soil or plant tissues was extracted in 5:1 *v*/*v* (HNO_3_/H_2_O) solution in an oven. Macro-minerals and trace element concentrations in the clear digestate were determined by mass spectroscopy (ICP-MS, Bremen, Germany) according to Hamad et al. [36].

### 2.10. Enzyme Activity Determination

Enzymes were extracted and their activities were measured according to described procedures. Assays were optimized to obtain linear time and protein-concentration dependence. Enzymes were extracted in potassium phosphate buffer (50 mM, pH 7.2) containing cysteine (5 mM), 5 mM EDTA, DTT (1 mM), insoluble polyvinyl pyrrolidone (PVP1%, (*w*/*v*)) and phenyl methyl sulfonyl fluoride (1 mM PMSF). Glutamine synthetase (GS, EC 6.3.1.2) was determined in Tris-acetate reaction buffer (Tris-acetate, 200 mM, pH 6.4) assayed following Robinson et al. [40] by monitoring of γ-glutamyl hydroxamate formation. Glutamine oxoglutarate aminotransferase (GOGAT, EC 2.6.1.53) was determined by following the glutamine-dependent NADH oxidation rate. The reaction mixture included of Tris-HCI (0.1 M, pH 7.5), glutamine, 2-oxoglutarte (0.33 M), and NADH (10 mM). Glutamate dehydrogenase (GDH, EC1.4.1.2) was determined in Tris–HCl buffer (100 mM, pH 8.1) containing α-ketoglutarate (20mM), (NH_4_)_2_SO_4_ (150 mM), NADH (0.2 mM), and MgCl_2_ (1 mM) by following the rate of 2-oxoglutarate-dependent NADH oxidation [40]. NADH-dependent nitrate reductase (NR, EC 1.7.1.1) activity was assayed in a potassium phosphate buffer (pH 7.6) mixture in a total volume of 2 cm^3^ consisting of 25 mM containing 100 mM KNO3, 1 mM NADH, and enzyme. The activity was assayed by measuring oxidation of NADH at A_340_ [41]. Protein concentrations were determined according to lowry methods [42] using bovine serum albumin (BSA) as the standard.

### 2.11. Assessment of Biological Activities

Fifty milligrams of legume seeds powder as well as 10 mg of actinomycetes isolates were extracted 3 times in 80% ethanol (*v*/*v*) using a MagNALyser (Roche, Vilvoorde, Belgium). Extracts were centrifuged at 14,000rpm for 30min at room temperature or 4 °C and the supernatants were used for the following assays.

#### 2.11.1. In Vitro Antioxidant Activity

DPPH and FRAP methods were used to determine the total antioxidant capacity of plant extracts. For extraction, 300 mg fresh tissue was homogenized in 3 mL 80% ice-cold ethanol. DPPH assay was performed by mixing the extract with DPPH dissolved in 95% ethanol (0.5 mL of 0.25 mM) [28]. After shaking and incubation at room temperature for 30 min, 2 mL of dd H_2_O was added and absorbance (517 nm) was determined using a micro-plate reader (Synergy Mx, Promega, Madison, WI, USA). 

Freshly prepared FRAP reagent was mixed with plant extracts in micro-plate and incubated (30 min, 37 °C), and the absorbance was measured at 593 nm [38]. A trolox calibration curve (0 to 650 µM) was used as a standard to calculate the antioxidant capacity.

#### 2.11.2. Antimicrobial Assay

Disc diffusion method [33] was used for assaying the potential antimicrobial activity of tissue ethanolic extracts against *Escherichia coli* and *Streptococcus* sp using a suspension (100 μL containing 1 × 10^8^ CFU/mL) of bacteria spread on Muller Hinton agar (SDA) [34]. Negative controls were prepared using the same solvent employed to dissolve extracts. Positive controls using amoxicillin (30 μg/disc), gentamycin (30 μg/disc), and streptomycin (30 μg/disc) were used in place of the tissue extract.

#### 2.11.3. Antiprotozoal Activity

In vitro assessment of the antiprotozoal activity of plant extracts against *Trypanosoma cruzi* was performed [35]. The viability of trypomastigotes of *Trypanosoma cruzi* (Tulahuen C4) was conducted by rat skeletal myoblasts (L-6 cells) seeded on the microtiter plates using chlorophenol red-β-D-galactopyranoside (CPRG)-Nonidet as a substrate. The microtiter plates were incubated for 24 h; then, they were incubated at 37 °C under 5% CO_2_ environment for 4 days and the developed color during the first 24 h was read photometrically at 540 nm.

### 2.12. Statistical Analyses

Data analyses (SPSS Inc., Chicago, IL, USA) and normality and homogeneity of variances (Kolmogorov–Smirnov test and Levene’s test, respectively) were performed. All the data were subjected to one-way analysis of variance (ANOVA). Duncan’s multiple range test (*p* < 0.05) was carried out as the post-hoc test for mean separations. The number of replicates for each vegetal species was three (*n* = 3). Principal component analysis (OriginLab 9, Northampton, MA, USA) and cluster analysis (Pearson distance metric, MultiExperiment Viewer (MeV)™ 4 software package, (La Jolla, CA, USA) of all results were performed.

## 3. Results

Actinomycetes are well known for their role in the nitrogen cycle and consequently on plant growth and productivity. Therefore, actinomycetes were isolated from the rhizosphere of nitrogen-fixing legume plants grown in the fields of Jouf region (Sakaka, Saudi Arabia) to study their effects on the nitrogen availability and crop yield of the legume plants.

### 3.1. Characterization of the Isolated Actinomycetes

Morphological investigations indicated that all the isolates have aerial mycelia, long spiral, or rectiflexible spore chains and some showed the ability to produce diffusible pigments (Appendix A). The isolates also showed extensively branched mycelia and coiled spore chains similar to the members of the genus *Streptomyces* (Figure 1). Moreover, we also characterized the actinomycete isolates by testing their capabilities to degrade some substrates and utilize different C or N sources (Appendix A).

### 3.2. Selection of the Potent Biologically Active Actinomycete Isolates

The isolated actinomycetes were screened for their total flavonoid and phenols contents, as well as their antioxidant, antiprotozoal, and antibacterial activities. The selection of the working biologically active isolates depended upon their total contents of the bioactive secondary compounds as well as their high biological activities, as indicated by their antimicrobial and antiprotozoal activities. Isolates 2, 8, 12, and 15 showed the highest content of the tested bioactive compounds and had the highest biological activities (Figure 2A). Moreover, the data showed a positive correlation between the flavonoid content of the actinomycete isolates and their biological activities such as antibacterial, antiprotozoal, and antioxidant capacities. The flavonoid profile was evaluated in the four selected actinomycete isolates (Figure 2B) as an indicator of their biological impact on the associated legume plant species.

### 3.3. The Four Selected Biologically Active Isolates belong to the Genus Streptomyces

The phylogenetic analysis confirmed that the four selected biologically active actinomycete strains belong to the genus *Streptomyces* (Figure 3). The phylogenetic tree revealed that the two isolates 2 and 15 are closely related to each other in one clade, while isolates 8 and 12 are in different clades in the tree. The tree also indicated that isolate 12 most likely represents a new species of the genus *Streptomyces*, which needs further taxonomic work to be confirmed.

### 3.4. Actinomycetes Improved Soil Fertility

The richness of the isolates 2, 8, 12, and 15 with bioactive secondary metabolites and their putative antimicrobial and antiprotozoal capacity prompted us to investigate the effect of the above-mentioned isolates on soil health and fertility. Our results show that the mineral content (Ca, P, K, Mg, Zn, and Cu), total phenolics, and organic matter were increased in soils after enriching them with the selected actinomycetes isolates (Appendix A). Moreover, the nitrogen content in the actinomycetes-enriched soils increased as compared to the control non-enriched soils (Table 1).

### 3.5. Nitrogen Availability and Metabolism

The improved soil chemical properties under the effect of the selected isolates of actinomycetes brought us to question how these changes affected the nitrogen and carbon content in the selected legumes (soybean, kidney bean, chickpea, lentil, and pea). The increased N and C content of soils after their enrichment with actinomycetes was reflected as a comparable increase in nitrogen and carbon content of plant tissues (roots and shoots) as well as seeds (Table 1). All selected isolates increased the nitrogen content in soil and isolate 12 and 15 caused the highest increase, while isolate 8 was the least effective. Similarly, legumes treated with isolates 2, 12, and 15 showed a considerable increase in nitrogen content in their roots, shoots, and seeds, while those treated with isolate 8 showed the least change in their nitrogen content.

To study the effect of the selected isolates on nitrogen metabolism in legume root and shoot, the activities of several N metabolism-related enzymes were investigated (Table 2). The activity of nitrate reductase (NR) in shoots and roots (but to a lesser extent) of any of the tested legume increased significantly as compared to untreated controls. GS activity increased several folds in the roots of treated plants and a slight but significant increase could be observed in their shoots. A similar trend of change, but to a lesser extent, could be observed for GOGAT activities. GDH activity of tested treated legume plants increased in the shoots, and to a lesser extent in roots as compared with their control untreated plants.

### 3.6. Improved Seed Yield and Chemical Composition Were Correlated with Increased Photosynthesis

To investigate whether the observed higher content of N and C in soils and tissues, as well as improved nitrogen metabolism, was correlated with improved photosynthesis, the content of chlorophyll and the rate of photosynthesis of the selected legumes was measured in control and actinomycetes-inoculated plants (Table 3). The different plants responded differently to the same isolate. All tested legumes showed increased chlorophyll (a and b) content and photosynthetic rates by all tested actinomycetes isolates. However, isolates 12 and 15 were the most potent organisms compared to their control untreated plants. Moreover, the primary metabolite (total proteins, lipids, and carbohydrates) content increased in the seeds of soybean, kidney bean, chickpea, lentil, and pea plants that were grown in soils enriched with any of the selected isolates (Figure 2C). All the assayed metabolites increased by all the selected isolates, especially isolates 12 and 15. Isolate 12 caused the most pronounced increase in protein content of seeds of all the tested legumes. The seeds of soybean showed the most increase of all the metabolites by all the isolates, especially isolate 12. In addition to increased photosynthetic rate, actinomycetes increased the soil content of essential micro- and macro-elements (Ca, P, K, Mg, Zn, and Cu) and enriched the soil with organic matter (Appendix A). Additionally, the selected actinomycete isolates generally improved the weight of the seeds in all of the legumes used in the experiment (Table 4). However, the response of the different legumes was different according to the different isolates treatments. The dry weight of lentil seeds increased by 300% compared to control by isolates 15, while that of chickpea seeds was not responsive to this isolate. The dry weight of kidney bean seeds did not increase in response to actinomycete treatments, except for isolate 15.

### 3.7. Amino Acid Composition

The profile of essential and non-essential amino acids in the seeds of actinomycete-treated legumes was investigated (Figure 3). Legumes responded differentially to the treatment with actinomycetes; for instance, pea seeds showed the highest increase in amino acids among all the tested legume seeds by all of the tested isolates. Kidney bean and chickpea seeds did not show a significant change in their amino acid profile under the influence of isolates 2, 8, and 12. The seeds of soybean showed an enhanced content of only Orn, His, Asp, Gln, and Tyr under the influence of 2, 12, and 15 and to a lesser extent by isolate 8.

### 3.8. Organic Acid Composition

Changes in the organic acids profile in different legume seeds by enriching the soil with all isolates of actinomycetes are shown in Figure 3. Isolates 12 and 15 caused the most dramatic increase in different organic acids content in all tested legume seeds as compared to control untreated plants, lentil and pea seeds being the most responsive ones. A slight increase and no significant changes in the organic acids in different legume seeds could be observed under the effect of isolates 2 and 8, respectively. Our results, however, show no lowering of soil pH value upon application of actinomycetes, so even if organic acids are secreted by the applied actinomycetes, they do not accumulate in the soil (Appendix A).

### 3.9. Fatty Acid Composition

Alterations in the lipid profile could dramatically affect the nutritional value of crops. The fatty acids profile of the selected leguminous seeds under study was monitored (Figure 3). Our results revealed that fatty acids compositions in seeds of legumes grown in soils enriched with the actinomycete isolates 12 and 15 substantially increased in all tested legume seeds. Isolate 2 was less effective and no significant change could be observed in seeds of plants treated with isolate 8. Tetracosanoic, hexadecenoic, and hexacosanoic acids were least affected in all seeds by any of the tested actinomycetes isolates. No specific pattern of change could be recognized, however, for saturated vs. unsaturated or long vs. short fatty acids under the effect of different actinomycetes isolates.

### 3.10. Actinomycetes Enhanced the Nutritional Value of the Treated Legumes

Overall, the treated legumes showed increased mineral, vitamin, and antioxidant levels in their seeds (Figure 4). For instance, the content of essential and non-essential mineral elements (Fe, Na, Cu, Cd, Zn, Mn, Ca, K, Mg, and/ or P) in seeds of all the tested legume seeds was increased by all isolates of actinomycetes. P content, however, was decreased by isolate 15 treatment.

At vitamins level, beta and gamma tocopherols, as well as cryptoxanthin, increased slightly in all seeds and the highest increase was in soybean seeds. Alpha-tocopherol and phylloquinone increased in all seeds by all strains. On the other hand, thiamine did not show any significant change in all seeds by any of the isolates. A remarkable increase in tocopherols and antioxidant content of seeds of all tested legumes can be observed under the effect of the different isolates of actinomycetes, especially isolates 12 and 15. Similarly, the flavonoid and phenolic content increased in seeds of all tested legumes, notably when treated with isolates 12 and 15 (Figure 4).

## 4. Discussion

With the aim of identifying potent actinomycetes strains which could be used to improve the nitrogen availability and metabolism in legumes, we isolated 17 isolates from legume cultivated agricultural fields in Al-Jouf region (Saudi Arabia) that potentially improve growth of plants and nutritional value of seeds of economically important leguminous crops (soybean, kidney bean, chickpea, lentil, and pea).

The cultural, morphological, and biochemical characteristics of the isolates indicate that the isolates belong mainly to the genus *Streptomyces* [43]. Out of the identified isolates, we identified 4 potent biologically active isolates (No. 2, 8, 12, and 15) based on their content of phenolics, their antioxidant capacities (FRAP and DPPH) as well as their antibacterial and antiprotozoal capacities. The capacity of actinomycetes to promote plant growth and protection against diseases have been found to correlate with their content of biologically active compounds like phenols and flavonoids as well as their antioxidant capacities [22]. Many actinomycetes can produce secondary metabolites when they grow in the rhizosphere of specific plants and these secondary metabolites play a role in plant protection against microbial diseases [10,44]. *Nocardiopsis alba* isolated from mangroves soil in Nellore region in India has been shown to produce secondary metabolites that show antioxidant and biological activities [45]. Moreover, different isolates of mycorrhizal-associated streptomycetes have been shown to produce secondary metabolites, such as the antibiotics cycloheximide and actiphenol that have shown potential inhibitory effects on several fungal and bacterial plant pathogens [46]. Notably, as isolates 12 and 15 showed the highest content of bioactive molecules and bioactivity; they also have the strongest effect on the tested legume plants. A similar correlation between bioactivity and richness of actinomycetes and improved metabolism and yield of cereals (wheat, oat, barley, maize, and sorghum) has been reported [22].

The antioxidant capacity and richness of the selected isolates with bioactive compounds can be correlated to the improved soil health and fertility, as indicated by the improved soil N, organic matter contents, soil nutrients availability as well as its content of phenolics and flavonoids. Similarly, amendment of cereal-grown soils with biologically active actinomycetes isolates improved their physical and chemical characteristics [22]. This is because actinomycetes are involved in organic matter cycling, fixation of atmospheric nitrogen, solubilization of phosphate, and suppression of plant pathogens [2,16,47]. Moreover, the observed increased soil content of elements after enrichment with actinomycetes could be explained by the chelating properties of the phenolics and flavonoids secreted by the actinomycetes. These chelating properties of phenolics and flavonoids have been previously described [48,49]. The improved soil characteristics and the biological activity of the selected actinomycetes isolates were reflected as improved legume growth and productivity. The improved chlorophyll content and photosynthesis of legumes grown in soils enriched with biologically active actinomycetes could be a result of the increased availablility of elements like N, P, and Mg that are essential for chlorophyll biosynthesis, and to the functionality of the photosynthetic machinary. *Micromonospora* and *Streptomyces* have phosphate solubilizing properties; they could improve the growth of wheat and kidney bean plants through their ability to solubilize phosphate, which in turn increased the growth, photosynthetic efficiency, and gain dry weight [11,16]. The improved photosynthesis consequently could have resulted in improved primary metabolism, as indicated by the improved seeds’ weight and their content of total carbohydrates, proteins, and lipids. Similarly, a remarkable increase in root, shoot growth, and biomass accumulation as well as N and P content in sweet corn plants after their inoculation with beneficial microorganisms including actinomycetes was reported [50]. Moreover, the improved photosynthesis in actinomycete treated plants resulted in an increased level of primary metabolites, as presented by total amounts of carbohydrates, proteins, and lipids. Moreover, the improved content of primary metabolites finally resulted in an increase in the grains’ weight. Likewise, two isolates of actinomycetes improved the growth and chlorophyll content of onion plants and conferred protection against postharvest rot diseases of bulbs [4]. It has been reported that actinomycete isolates improved the organic matter content as well as the micro- and macro-element content of three types of problematic soils [51]. A comparable effect of actinomycetes isolates on improving the tiller and panicle numbers, as well as enhanced grain number and weight in rice have also been tested by Gopalakrishnan et al. [52]. In addition to improved photosynthetic rates in the tested actinomycete-treated legume plants, the N availability and its improved metabolism could have also contributed to the abundance of the primary metabolites (notably proteins) compared to their levels in control untreated plants. A remarkable increase in root, shoot growth, and biomass accumulation as well as N and P content in sweet corn plants after their inoculation with beneficial microorganisms including actinomycetes was reported [50]. Some reports have shown positive effects of actinomycetes on the nitrogen availability and growth of different plants. For example, soybean soil inoculated with different actinomycetes (*Streptomycetes*, *Nocardia*, *Nonomurea*, and *Actinomadura*) resulted in increased plant growth, total dry weight, and mineral composition [53]. Actinomycete co-inoculation with *Bradyrhizobium japonicum* further stimulated soybean growth through increasing nitrogenase activity of root nodules, nutrients uptake, and seeds’ weight at harvest compared to plants inoculated only with *B. japonicum* [53,54]. Similarly, the endophytic nitrogen bacterium *Enterobacter* improved the growth and seed yield of the *Jatropha* plant [55]. The enhanced activities of the nitrogen metabolism (NR, GS, GDH, and GOGAT) especially under the effect of isolates 12 and 15 highlights the potentiality of actinomycetes in positively altering the growth and metabolism of plants and give mechanistic insight on how these positive effects are implemented. A direct relation between N availability and promoted growth has been reported in plants, and the underlying mechanisms include enhanced activities of nitrogen metabolizing enzymes. In 3 finger millet genotypes, higher nitrogen use efficiency was linked to the higher activities of GDH, GS, NR, and GOGAT enzymes, especially during the vegetative growth [56]. Although the different isolated bioactive actinomycetes improved the growth and yield of the tested legumes through improving soil characters, mineral availability, and nitrogen metabolism, the different tested plants showed different responses to the different actinomycetes isolates in terms of the plants growth and yield. These differences could have resulted from observed differences in metabolite (amino acids, fatty acids, and organic acids) profiles between the plants.

These differences in metabolism and yield between tested plants may be due to the different requirements of these plants for minerals and soil environment. These differences are due to differences in genetic composition and evolutionary/breeding history of plants. Such differences in mineral and soil requirements of different plants also affect their interaction with soil biota and their responses to abiotic stresses [57,58,59]. The difference in interactions between different plants roots and actinomycetes could also contribute to the observed variations in plant responses. For example, some amino acids (e.g., glutamic acid, alanine, valine, and lysine) were released by *Streptomycetes* into their rhizosphere [60]. The released amino acids could be differently absorbed by the plant roots, so they improve nitrogen metabolism in plants and the amino acid content of the seeds to varying degrees. The absorption of organic nitrogen such as amino acids by roots of plants has been reported. Roots of sugarcane in tissue culture supplied with five amino acids, that normally exist in soils where sugarcane grows, were able to absorb them and the plants accumulated biomass similar to those plants supplied with inorganic nitrogen [61]. Similarly, organic acids like pyruvate, lactate, oxalate, malate, and succiniate are produced and released into the soil by *Streptomycetes* spp. and affect the soil abiotic and biotic environment, mineral availability, and root growth and ion absorption by roots [47,62,63]. The increase in organic acid content and the differences in their profile in seeds could be a result of the higher carbohydrates content with the contaminant increase in the concentration and relative availability of tricarboxylic acid cycle intermediates compared to their control untreated counterparts. Similarly, Vyas and Gulati [64] found that the treatment of maize seeds with fluorescent *Pseudomonas* significantly improved the accumulation of organic acids in maize plants. Moreover, actinomycetes were involved in lytic enzymes and organic acid production in soils that may enhance soil fertility [65]. In this context, organic acid accumulation might lead to soil biological buffering by suppressing growth of plant pathogens in the rhizosphere [2]. Concerning the plants, although the presence of organic acids did not impact the grains’ nutritive value, it could prolong their storage life through microorganisms’ growth inhibition.

The improved seed content of different metabolites is directly linked to their nutritional values. The legume seeds contained a profitable composition of polyunsaturated fatty acids. Polyunsaturated fatty acids function as main nutrients, constituents of cell membranes, and precursors of various signal molecules [66]. This improvement in fatty acid composition provides the seeds of our tested plants with more nutritive value [67]. Actinomycetes isolated from pine tree roots could produce and release B-type vitamins into the soil [68], which can be taken up by plant roots [69].

## 5. Conclusions

Overall, our results show that the grain yield enhancement of plants grown in soils inoculated with biologically active actinomycetes may be a result of the actinomycete improved soil chemical properties of soils. In this regard, actinomycetes such as enhanced organic matter and nitrogen content as well as essential macro and micro-elements, which in turn improved plant growth, carbon metabolism and allocation, and finally improved grain quality and yield. The enhanced nitrogen content after actinomycete application mimics the application of N fertilizer, so actinomycetes could be a self-sustainable, eco-friendly nitrogen fertilizer. Moreover, the rich phenolic and flavonoid content of the applied actinomycetes could have made more nutrient elements available to plant roots and could also protect plants against pathogens. However, the actinomycetes’ direct impact on plant metabolism and productivity and the characterization of their products need deeper investigations. Our results are also the first report that shows the direct positive influence of biologically active actinomycetes on the nitrogen metabolism of plants and ultimately higher yield quantity and better nutritional quality. The results also show a promising environment-friendly alternative of agrochemicals that the world urgently needs.

## Figures and Tables

**Figure 1 biomolecules-10-01675-f001:**
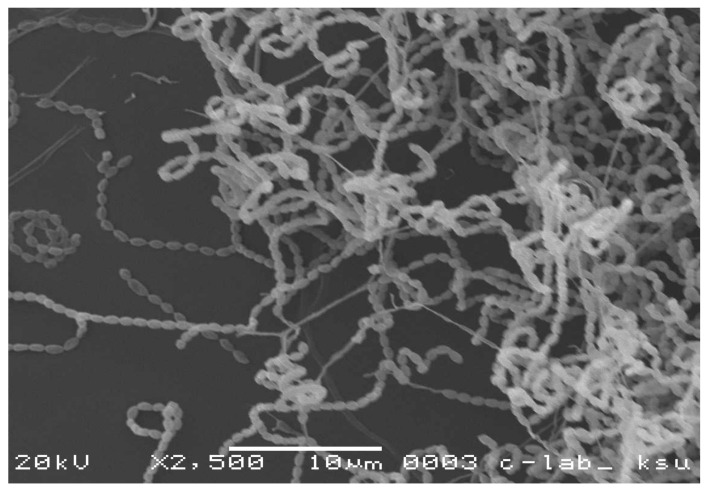
Scanning electron micrograph of actinomycete isolate 2 as a representative photo of the Streptomyces isolates used in this study (2500×).

**Figure 2 biomolecules-10-01675-f002:**
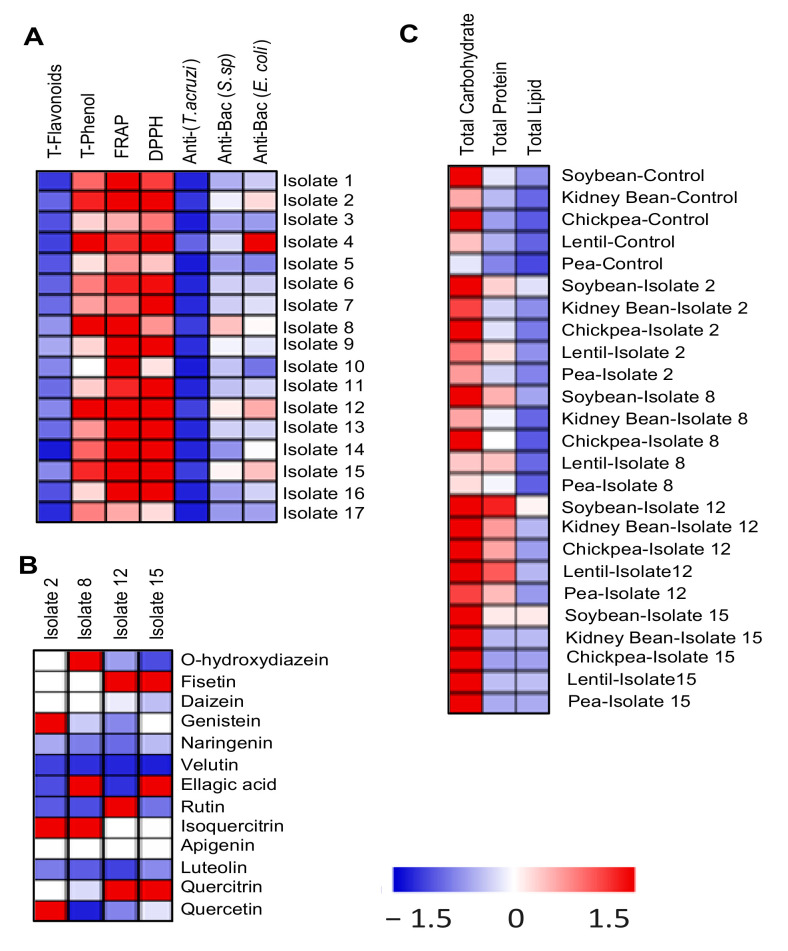
(**A**) Heatmap showing the results of screening of the different actinomycete isolates isolated from local legume fields for their biological activity as indicated by their total flavonoids and phenolics content, total antioxidant capacity measured by FRAP and DPPH assays, as well as their antiprotozoal activities. (**B**) The flavonoid profile in the four selected actinomycete isolates. (**C**) The changes in total carbohydrates, proteins, and lipids in grains of the selected legumes upon soil enrichment with biologically active actinomycetes.

**Figure 3 biomolecules-10-01675-f003:**
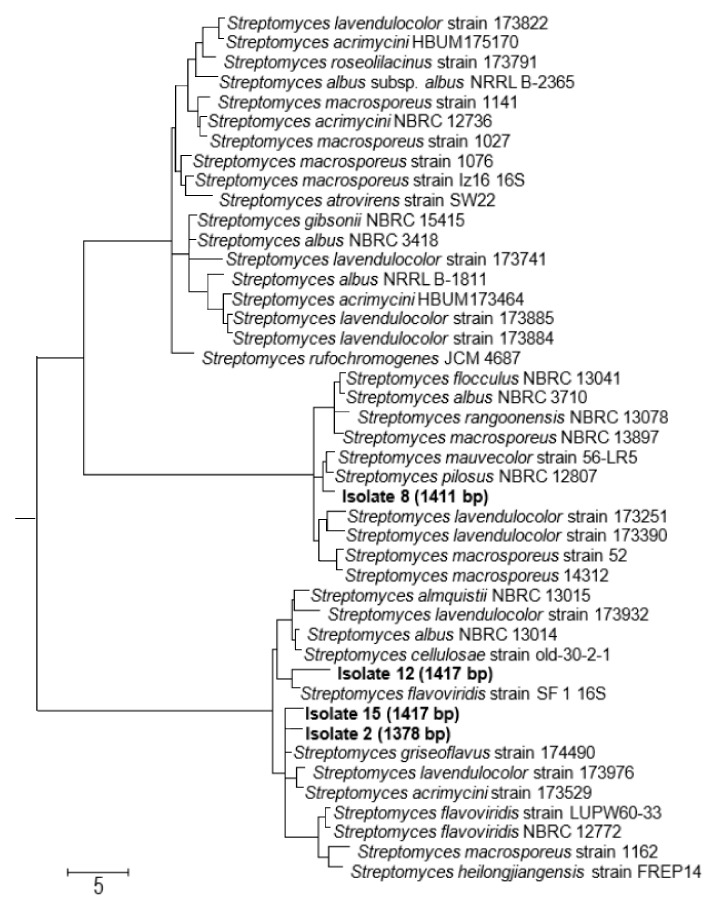
A neighbour joining tree based on 16S rRNA gene sequences showing the relationships between the four selected isolates (**bold font**) and their closely related members of the genus *Streptomyces*. Numbers at the nodes are percentage bootstrap values based on 1000 resampled datasets; only values above 50% are given. Bar, 0.05 substitutions per nucleotide position.

**Figure 4 biomolecules-10-01675-f004:**
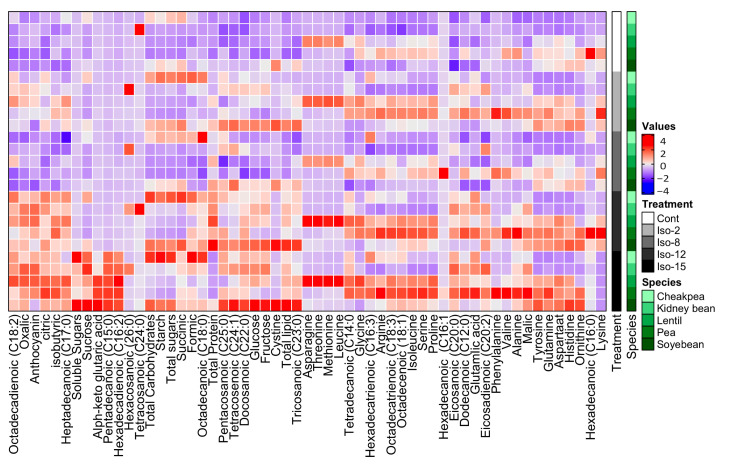
Heatmap showing the changes in the profile of amino acids, carbohydrates, organic acids, and lipids in seeds of soybean, kidney bean, chickpea, lentil, and pea under the effect of soil enrichment with biologically active actinomycete isolates.

**Table 1 biomolecules-10-01675-t001:** Changes in nitrogen and carbon content (mg/g) of shoots, roots, seeds as well as soil of soybean, kidney bean, chickpea, lentil, and pea under control (cont.) and after enrichment with the biologically active actinomycete isolates. Values are the average of three individual replicates (means ± S.D). Different letters represent significant differences between the treatments in each crop (Duncan test; *p* < 0.05; *n* = 4).

	Nitrogen				Carbon		
Plant-Isolate	Root	Shoot	Seed	Soil	Root	Shoot	Seed
Soybean-Cont	34.8 ± 0.8 ^a^	51.5 ± 1.1 ^a^	29.1± 0.8 ^a^	0.4 ± 0.01 ^a^	239.5 ± 9.60 ^a^	557.0 ± 90.5 ^a^	104.8 ± 0.40 ^a^
Soybean-I2	51.1 ± 3.5 ^a^	65.8 ± 1.5 ^b^	41.2 ± 2.1 ^b^	0.6 ± 0.01 ^b^	329.6 ± 13.0 ^b^	374.7 ± 0.00 ^a^	131.2 ± 5.40 ^b^
Soybean-I8	52.4 ± 4.0 ^a^	48.1 ± 1.1 ^a^	38.3 ± 2.1 ^a^	0.4 ± 0.20 ^a^	281.0 ± 11.3 ^a^	390.9 ± 0.00 ^a^	136.2 ± 1.70 ^b^
Soybean-I12	55.7 ± 1.6 ^b^	71.4 ± 1.6 ^b^	50.9 ± 0.5 ^b^	1.0 ± 0.20 ^a^	404.3 ± 16.0 ^b^	426.1 ± 0.00 ^a^	159.5 ± 5.20 ^b^
Soybean-I15	66.8 ± 3.7 ^b^	86.7 ± 1.9 ^b^	55.8 ± 0.5 ^b^	1.0 ± 0.10 ^a^	434.3 ± 17.0 ^b^	425.1 ± 0.00 ^a^	112.0 ± 20.7 ^a^
Kidney Bean-Cont	48.3 ± 3.9 ^a^	53.5 ± 2.9 ^a^	37.4 ± 1.5 ^a^	0.4 ± 0.03 ^a^	292.2 ± 1.10 ^a^	349.8 ± 8.10 ^a^	158.7 ± 1.40 ^a^
Kidney Bean-I2	58.9 ± 7.3 ^a^	55.8 ± 3.0 ^a^	38.8 ± 0.2 ^a^	0.5 ± 0.00 ^a^	331.0 ± 1.30 ^b^	421.4 ± 0.00 ^a^	161.3 ± 2.90 ^a^
Kidney Bean-I8	51.6 ± 7.3 ^a^	40.8 ± 2.2 ^a^	37.9 ± 2.6 ^a^	0.5 ± 0.00 ^a^	290.9 ± 1.00 ^a^	392.3 ± 0.00 ^a^	219.1 ± 1.30 ^a^
Kidney Bean-I12	62.1 ± 9.2 ^a^	57.9 ± 3.2 ^a^	38.5 ± 0.2 ^a^	0.6 ± 0.00 ^b^	327.5 ± 1.00 ^b^	433.9 ± 0.00 ^a^	160.8 ± 3.60 ^a^
Kidney Bean-I15	61.3 ± 6.4 ^a^	73.5 ± 4.0 ^a^	33.6 ± 0.3 ^a^	0.7 ± 0.00 ^b^	262.3 ± 1.00 ^b^	404.3 ± 0.00 ^a^	235.5 ± 6.50 ^a^
Chickpea-Cont	44.4 ± 1.4 ^a^	54.0 ± 2.9 ^a^	36.2 ± 0.6 ^a^	0.3 ± 0.10 ^a^	295.0 ± 1.20 ^a^	236.9 ± 38.0 ^a^	170.8 ± 2.80 ^a^
Chickpea-I2	56.2 ± 5.1 ^a^	83.0 ± 4.5 ^b^	37.5 ± 0.3 ^a^	0.7 ± 0.01 ^b^	292.0 ± 1.10 ^a^	92.8 ± 0.00 ^a^	219.0 ± 2.60 ^a^
Chickpea-I8	48.8 ± 3.8 ^a^	60.7 ± 3.3 ^a^	37.4 ± 1.4 ^a^	0.6 ± 0.01 ^a^	285.0 ± 1.10 ^b^	496.9 ± 0.00 ^a^	232.1 ± 1.50 ^a^
Chickpea-I12	41.7 ± 0.5 ^a^	61.9 ± 3.4 ^a^	34.8 ± 0.2 ^a^	0.6 ± 0.00 ^b^	285.8 ± 1.00 ^b^	158.9 ± 0.00 ^a^	237.6 ± 3.50 ^a^
Chickpea-I15	51.9 ± 1.0 ^a^	109.4 ± 6.0 ^b^	40.9 ± 0.5 ^b^	0.9 ± 0.10 ^b^	299.6 ± 1.00 ^a^	105.3 ± 0.00 ^a^	197.6 ± 1.70 ^b^
Lentil-Cont	37.0 ± 0.9 ^a^	54.5 ± 1.2 ^a^	30.9 ± 0.9 ^a^	0.3 ± 0.10 ^a^	254.7 ± 10.0 ^a^	304.7 ± 34.0 ^a^	357.7 ± 2.10 ^a^
Lentil-I2	41.3 ± 2.2 ^a^	57.8 ± 1.3 ^a^	41.9 ± 1.3 ^b^	0.6 ± 0.01 ^a^	361.2 ± 14.0 ^b^	2172.8 ± 0.00 ^b^	458.0 ± 16.8 ^b^
Lentil-I8	35.0 ± 0.6 ^a^	53.1 ± 1.2 ^a^	31.7 ± 0.9 ^a^	0.5 ± 0.00 ^a^	264.0 ± 10.6 ^a^	2267.2 ± 0.00 ^b^	480.2 ± 9.60 ^b^
Lentil-I12	57.7 ± 1.6 ^b^	65.6 ± 1.5 ^b^	55.6 ± 1.8 ^b^	1.1 ± 0.20 ^a^	435.3 ± 18.0 ^b^	2497.7 ± 0.00 ^b^	533.0 ± 19.0 ^b^
Lentil-I15	63.1 ± 1.8 ^b^	71.2 ± 1.3 ^b^	60.8 ± 2.0 ^b^	0.7 ± 0.00 ^b^	476.0 ± 19.1 ^b^	2465.2 ± 0.00 ^b^	392.0 ± 60.5 ^a^
Pea-Cont	32.0 ± 0.7 ^a^	52.8 ± 1.2 ^a^	26.3 ± 0.7 ^a^	0.3 ± 0.10 ^a^	210.5 ± 8.00 ^a^	435.3 ± 50.0 ^a^	158.7 ± 21.0 ^a^
Pea-I2	43.5 ± 3.0 ^a^	63.1 ± 1.4 ^b^	42.6 ± 1.3 ^b^	0.7 ± 0.00 ^b^	363.2 ± 15.0 ^b^	1497.3 ± 0.00 ^b^	337.2 ± 13.0 ^b^
Pea-I8	35.9 ± 0.7 ^a^	57.6 ± 1.3 ^a^	32.3 ± 0.9 ^b^	0.5 ± 0.10 ^a^	265.4 ± 10.0 ^a^	1562.3 ± 0.00 ^b^	293.2 ± 23.0 ^a^
Pea-I12	53.9 ± 1.4 ^b^	69.9 ± 1.6 ^b^	51.3 ± 1.6 ^b^	0.8 ± 0.10 ^b^	392.1 ± 16.0 ^b^	728.1 ± 0.00 ^a^	268.3 ± 9.60 ^a^
Pea-I15	75.9 ± 1.8 ^b^	72.7 ± 1.6 ^b^	57.4 ± 3.1 ^b^	1.1 ± 0.20 ^a^	478.6 ± 19.0 ^b^	1698.9 ± 0.00 ^b^	305.0 ± 41.3 ^a^

**Table 2 biomolecules-10-01675-t002:** Changes in the activity of nitrogen metabolizing enzymes glutamate dehydrogenase (GDH), glutamine synthetase (GS), nitrate reductase (NR), and glutamate synthase (GOGAT) expressed as units/mg protein in the shoots and roots of soybean, kidney bean, chickpea, lentil, and pea under control (cont.) and after enrichment with biologically active actinomycete isolates. Values are the average of three individual replicates (means ± S.D). Different letters represent significant differences between the treatments in each crop (Duncan test; *p* < 0.05; *n* = 4).

	GDH		GS		NR		GOGAT	
Plant-Isolate	Root	Shoot	Root	Shoot	Root	Shoot	Root	Shoot
Soybean-Cont	0.97 ± 0.09 ^a^	0.47 ± 0.03 ^a^	1.99 ± 0.49 ^a^	0.95 ± 0.21 ^a^	0.84 ± 0.01 ^a^	0.22 ± 0.01 ^a^	1.82 ± 0.06 ^a^	0.59 ± 0.05 ^a^
Soybean-I2	0.91 ± 0.08 ^a^	0.49 ± 0.08 ^a^	6.56 ± 0.39 ^b^	1.45 ± 0.04 ^a^	1.53 ± 0.15 ^a^	0.20 ± 0.02 ^a^	2.39 ± 0.04 ^b^	0.63 ± 0.05 ^a^
Soybean-I8	0.93 ± 0.09 ^a^	0.80 ± 0.11 ^a^	6.70 ± 0.39 ^b^	2.32 ± 0.08 ^b^	1.09 ± 0.18 ^a^	0.27 ± 0.01 ^a^	1.53 ± 0.02 ^b^	0.64 ± 0.05 ^a^
Soybean-I12	1.20 ± 0.11 ^a^	0.69 ± 0.08 ^a^	8.31 ± 0.49 ^b^	1.63 ± 0.09 ^a^	1.56 ± 0.02 ^b^	0.32 ± 0.02 ^b^	3.22 ± 0.10 ^b^	0.75 ± 0.06 ^a^
Soybean-I15	0.97 ± 0.09 ^a^	0.97 ± 0.18 ^a^	6.99 ± 0.41 ^b^	1.17 ± 0.11 ^a^	1.87 ± 0.08 ^b^	0.28 ± 0.01 ^a^	3.80 ± 0.25 ^b^	0.67 ± 0.05 ^a^
Kidney Bean-Cont	0.92 ± 0.05 ^a^	0.51 ± 0.00 ^a^	2.63 ± 0.21 ^a^	1.67 ± 0.24 ^a^	0.05 ± 0.01 ^a^	0.13 ± 0.02 ^a^	1.59 ± 0.23 ^a^	6.91 ± 0.65 ^a^
Kidney Bean-I2	1.07 ± 0.08 ^a^	0.91 ± 0.13 ^a^	7.56 ± 0.45 ^b^	1.69 ± 0.11 ^a^	0.15 ± 0.00 ^b^	0.20 ± 0.02 ^a^	2.54 ± 0.04 ^a^	9.83 ± 0.76 ^a^
Kidney Bean-I8	1.10 ± 0.08 ^a^	1.43 ± 0.23 ^a^	7.72 ± 0.45 ^b^	2.81 ± 0.24 ^a^	0.09 ± 0.01 ^a^	0.31 ± 0.02 ^b^	1.62 ± 0.08 ^a^	10.05 ± 0.78 ^a^
Kidney Bean-I12	1.36 ± 0.10 ^a^	1.00 ± 0.12 ^a^	9.66 ± 0.57 ^b^	2.50 ± 0.10 ^a^	0.17 ± 0.00 ^b^	0.41 ± 0.02 ^b^	3.38 ± 0.06 ^b^	10.05 ± 0.78 ^a^
Kidney Bean-I15	1.14 ± 0.09 ^a^	0.80 ± 0.02 ^b^	8.05 ± 0.47 ^b^	1.73 ± 0.06 ^a^	0.20 ± 0.00 ^b^	0.32 ± 0.02 ^b^	3.38 ± 0.05 ^b^	10.48 ± 0.81 ^a^
Chickpea-Cont	1.07 ± 0.08 ^a^	0.49 ± 0.01 ^a^	2.37 ± 0.49 ^a^	1.33 ± 0.13 ^a^	0.15 ± 0.05 ^a^	0.15 ± 0.01 ^a^	0.85 ± 0.15 ^a^	0.76 ± 0.06 ^a^
Chickpea-I2	0.90 ± 0.07 ^a^	0.73 ± 0.17 ^a^	6.03 ± 0.36 ^b^	1.15 ± 0.04 ^a^	0.52 ± 0.01 ^b^	0.13 ± 0.01 ^a^	1.83 ± 0.03 ^b^	0.93 ± 0.07 ^a^
Chickpea-I8	0.92 ± 0.07 ^a^	1.03 ± 0.12 ^a^	6.16 ± 0.36 ^b^	1.67 ± 0.05 ^a^	0.36 ± 0.02 ^a^	0.14 ± 0.01 ^a^	1.27 ± 0.04 ^a^	0.95 ± 0.07 ^a^
Chickpea-I12	1.29 ± 0.10 ^a^	0.80 ± 0.10 ^a^	7.90 ± 0.46 ^b^	1.95 ± 0.08 ^a^	0.49 ± 0.01 ^b^	0.18 ± 0.01 ^a^	2.00 ± 0.03 ^b^	1.00 ± 0.08 ^a^
Chickpea-I15	0.96 ± 0.07 ^a^	0.79 ± 0.09 ^a^	6.42 ± 0.38 ^b^	0.97 ± 0.03 ^a^	0.70 ± 0.01 ^b^	0.14 ± 0.01 ^a^	2.48 ± 0.04 ^b^	0.99 ± 0.08 ^a^
Lentil-Cont	0.88 ± 0.05 ^a^	0.47 ± 0.02 ^a^	2.07 ± 0.29 ^a^	1.31 ± 0.07 ^a^	0.39 ± 0.07 ^a^	0.16 ± 0.01 ^a^	0.41 ± 0.01 ^a^	1.43 ± 0.36 ^a^
Lentil-I2	1.05 ± 0.09 ^a^	0.91 ± 0.07 ^b^	5.08 ± 0.30 ^b^	1.06 ± 0.08 ^a^	1.43 ± 0.18 ^b^	0.19 ± 0.04 ^a^	0.60 ± 0.03 ^b^	1.23 ± 0.31 ^a^
Lentil-I8	1.07 ± 0.09 ^a^	0.96 ± 0.18 ^a^	5.19 ± 0.31 ^b^	1.42 ± 0.15 ^a^	0.72 ± 0.08 ^a^	0.22 ± 0.03 ^a^	0.50 ± 0.04 ^a^	3.25 ± 0.23 ^a^
Lentil-I12	1.31 ± 0.11 ^a^	0.58 ± 0.07 ^a^	6.28 ± 0.37 ^b^	1.22 ± 0.07 ^a^	1.73 ± 0.20 ^b^	0.28 ± 0.04 ^a^	0.73 ± 0.02 ^b^	1.74 ± 0.44 ^a^
Lentil-I15	1.12 ± 0.09 ^a^	0.80 ± 0.05 ^b^	5.41 ± 0.32 ^b^	0.93 ± 0.04 ^a^	1.34 ± 0.21 ^a^	0.23 ± 0.03 ^a^	0.78 ± 0.01 ^b^	1.43 ± 0.31 ^a^
Pea-Cont	0.62 ± 0.09 ^a^	0.55 ± 0.02 ^a^	2.71 ± 0.44 ^a^	1.03 ± 0.09 ^a^	0.30 ± 0.07 ^a^	0.12 ± 0.01 ^a^	1.21 ± 0.11^a^	3.87 ± 0.39 ^a^
Pea-I2	1.27 ± 0.08 ^b^	0.89 ± 0.07 ^b^	6.37 ± 0.38 ^b^	1.22 ± 0.09 ^a^	1.33 ± 0.21 ^a^	0.20 ± 0.03 ^a^	4.81 ± 0.22 ^b^	6.70 ± 0.52 ^a^
Pea-I8	1.30 ± 0.09 ^b^	0.72 ± 0.03 ^a^	6.51 ± 0.38 ^b^	1.66 ± 0.17 ^a^	0.98 ± 0.19 ^a^	0.27 ± 0.01 ^b^	4.02 ± 0.32 ^b^	6.84 ± 0.53 ^a^
Pea-I12	1.47 ± 0.10 ^b^	0.60 ± 0.07 ^a^	6.76 ± 0.40 ^b^	1.25 ± 0.07 ^a^	1.98 ± 0.01 ^b^	0.29 ± 0.01 ^b^	5.70 ± 0.18 ^b^	9.55 ± 0.74 ^b^
Pea-I15	1.36 ± 0.09 ^b^	0.86 ± 0.04 ^b^	6.79 ± 0.40 ^b^	1.11 ± 0.04 ^a^	1.82 ± 0.35 ^a^	0.28 ± 0.01 ^b^	5.91 ± 0.10 ^b^	7.13 ± 0.55 ^b^

**Table 3 biomolecules-10-01675-t003:** Changes in photosynthetic rate (µmol CO_2_ m^−2^ s ^−1^) and chlorophyll a + b content (mg/g FW) in soybean, kidney bean, chickpea, lentil, and pea under control (cont.) and after enrichment with biologically active actinomycete isolates. Values are the average of three individual replicates (means ± S.D). Different letters represent significant differences between the treatments in each crop (Duncan test; *p* < 0.05; *n* = 4).

Plant-Isolate	Photosynthetic Rate	Ch a+b Content
Soybean-Cont	15.17 ± 1.20 ^a^	4.26 ± 0.20 ^a^
Soybean-I2	21.82 ± 1.20 ^a^	6.30 ± 0.91 ^a^
Soybean-I8	18.55 ± 3.30 ^b^	6.71 ± 0.20 ^ab^
Soybean-I12	26.68 ± 0.03 ^b^	8.87 ± 0.50 ^b^
Soybean-I15	22.30 ± 1.33 ^b^	4.48 ± 0.70 ^b^
Kidney Bean-Cont	17.93 ± 1.60 ^a^	4.92 ± 0.31 ^a^
Kidney Bean-I2	26.23 ± 2.03 ^a^	5.14 ± 0.42 ^a^
Kidney Bean-I8	19.01 ± 2.10 ^a^	5.55 ± 1.00 ^b^
Kidney Bean-I12	20.03 ± 4.25 ^a^	5.40 ± 1.10 ^b^
Kidney Bean-I15	16.00 ± 2.61 ^a^	5.14 ± 1.50 ^b^
Chickpea-Cont	21.26 ± 1.40 ^a^	3.87 ± 0.77 ^a^
Chickpea-I2	23.31 ± 2.43 ^b^	7.26 ± 0.86 ^b^
Chickpea-I8	19.29 ± 3.80 ^a^	4.13 ± 0.20 ^a^
Chickpea-I12	19.05 ± 1.54 ^a^	4.70 ± 0.80 ^ab^
Chickpea-I15	21.29 ± 2.10 ^a^	5.60 ± 1.70 ^b^
Lentil-Cont	18.11 ± 0.80 ^a^	4.76 ± 0.51 ^a^
Lentil-I2	31.27 ± 1.13 ^b^	7.44 ± 0.97 ^b^
Lentil-I8	22.11 ± 1.50 ^b^	6.41 ± 0.70 ^a^
Lentil-I12	30.28 ± 1.21 ^b^	8.23 ± 0.09 ^a^
Lentil-I15	38.97 ± 1.10 ^b^	9.30 ± 1.14 ^a^
Pea-Cont	16.09 ± 0.60 ^a^	5.02 ± 0.84 ^a^
Pea-I2	26.51 ± 1.10 ^b^	7.12 ± 1.30 ^b^
Pea-I8	21.54 ± 0.60 ^a^	6.34 ± 0.30 ^b^
Pea-I12	23.34 ± 2.21 ^b^	9.46 ± 0.91 ^a^
Pea-I15	37.40 ± 2.12 ^b^	9.10 ± 0.85 ^b^

**Table 4 biomolecules-10-01675-t004:** Seed yield measured as dry weight per 100 seed of soybean, kidney bean, chickpea, lentil, and pea under control (cont.) and after enrichment with biologically active actinomycete isolates. Values are the average of three individual replicates (means ± S.D). Different letters represent significant differences between the treatments in each crop (Duncan test; *p* < 0.05; *n* = 4).

Plant-Isolate	Seed Yield (DW/100 Seed)
Soybean-Cont.	15.77 ± 0.8 ^a^
Soybean-I2	20.18 ± 0.1 ^b^
Soybean-I8	27.01 ± 0.2 ^b^
Soybean-I12	13.68 ± 3.70 ^a^
Soybean-I15	19.86 ± 0.50 ^b^
Kidney Bean-Cont.	16.91 ± 0.10 ^a^
Kidney Bean-I2	17.60 ± 0.70 ^a^
Kidney Bean-I8	15.78 ± 0.90 ^a^
Kidney Bean-I12	14.14 ± 3.00 ^a^
Kidney Bean-I15	24.31 ± 17.0 ^a^
Chickpea-Cont.	23.37 ± 0.17 ^a^
Chickpea-I2	25.20 ± 1.90 ^a^
Chickpea-I8	32.35 ± 1.10 ^b^
Chickpea-I12	29.70 ± 0.41 ^b^
Chickpea-I15	20.03 ± 3.80 ^a^
Lentil-Cont.	2.10 ± 0.10 ^a^
Lentil-I2	4.28 ± 0.50 ^b^
Lentil-I8	3.01 ± 0.90 ^b^
Lentil-I12	5.66 ± 0.81 ^b^
Lentil-I15	5.90 ± 0.70 ^b^
Pea-Cont.	12.78 ± 1.10 ^a^
Pea-I2	19.65 ± 0.57 ^b^
Pea-I8	15.39 ± 2.00 ^a^
Pea-I12	14.19 ± 2.21 ^a^
Pea-I15	23.56 ± 2.00 ^b^

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
