# Peer review of "Actinomycetes Enrich Soil Rhizosphere and Improve Seed Quality as well as Productivity of Legumes by Boosting Nitrogen Availability and Metabolism"

_biomolecules, 2020, doi:10.3390/biom10121675_

Round 1

Reviewer 1 Report

Authors report results on the isolation and characterization of actinomycetes strains from rhizosphere soil of legume crops. Four isolates selected on the basis of in vitro bioactivity traits were applied on 5 legumes (i.e. soybean, kidney bean, chickpea, lentil, and pea) and the effects on soils and plants were evaluated. Soil enrichment with selected isolates increased soil nutrients and organic matter content improved soil microbial populations and photosynthetic reactions and yield of plants. Enrichment also improved soil nitrogen availability and legume tissue and seeds, which induced the activity of key nitrogen metabolizing enzymes.

INTRODUCTION

MATERIAL AND METHODS

This section should be deeply ameliorated in all its subsections. The experiments were described with few details and, in some cases, there are no references that could help readers to understand or repeat the experiment. Some examples:

L105 – sample preparation for scanning electron microscope analysis and working conditions are missing.

L109 – The ratio of broth culture/soil combination is missing. Authors should describe more in detail this.

L110 – How long after inoculation were soil used for sowing?

L111 – seed details are missing (e.g. variety, cultivar, company). Moreover, how was the seed surface sterilization carried out?

L128 – Which were the appropriate media?

L134-141 - Metabolite profiling were these analyses performed on broth cultures? I suggest to clearly present two separated sections: one on isolates screening and another on plant extracts. At the beginning of each section should be presented the extraction procedure/the way authors get the material to be investigated with the different assays.

L140-141 – Revise the sentence and provide more details about methods.

L154 – Revise the sentence.

L155 – I suggest to add immediately after section 2.10 title the details of plant tissues extraction. Please, provide more details about the homogenization procedure.

L179 – Which was the standard kit used for DNA isolation? Please, provide name and company details.

RESULTS AND DISCUSSION

The results are presented correctly, however, the not clear materials and methods section preclude to understand some findings. In general, the discussion is too short and is only limited to the comparison of some other authors findings. Authors should ameliorate the discussion; perhaps, a separated discussion section could help authors to ameliorate it.

L199-206 – Authors should discuss more in detail the implication of some of the characteristic of the isolates. Are these characteristic important for the plant growth-promoting abilities? Can these characteristics be taken into account for the utilization of these strains for agricultural applications?

L211-233 – According to the materials and methods section, the total flavonoid and phenolic contents, antioxidant, antiprotozoal and antibacterial activities were performed on plant extracts. In this section are presented results of the same assays as bioactivity screening of actinomycetes isolates. Authors should clarify the methods in materials and methods section and modify this section accordingly.  

L250-257 – The discussion of these findings should be ameliorated.

L458; L462 – grain/grains yield?

CONCLUSIONS

This section is appropriate; however, authors should ameliorate the previous sections to get the validity of the statements made.

MINOR ISSUES

L46 – “They can also biosynthesize…” It is not clear the subject of the sentence. Please, rephrase.

L52 – “biocontrol agents, and plant growth-promoting chemicals”

L91-94 – The sentence is too long and it is not grammatically correct. Please, rephrase.

Some format issues: L95, L197 – Title; L156 – italics; L140, L176 – Subscript;

L189 – substitute “cultivar” with “crop” or “vegetal species”.

Figure 2 quality should be ameliorated.

L301-307 – Authors start the sentence with “The positive effect of actinomycetes on nitrogen fixation by root nodules has been reported.”, however, only one study on actinomycetes has been reported.

Author Response

Reviewer#1

Authors report results on the isolation and characterization of actinomycetes strains from rhizosphere soil of legume crops. Four isolates selected on the basis of in vitro bioactivity traits were applied on 5 legumes (i.e. soybean, kidney bean, chickpea, lentil, and pea) and the effects on soils and plants were evaluated. Soil enrichment with selected isolates increased soil nutrients and organic matter content improved soil microbial populations and photosynthetic reactions and yield of plants. Enrichment also improved soil nitrogen availability and legume tissue and seeds, which induced the activity of key nitrogen metabolizing enzymes.

 INTRODUCTION

 MATERIAL AND METHODS

This section should be deeply ameliorated in all its subsections. The experiments were described with few details and, in some cases, there are no references that could help readers to understand or repeat the experiment. Some examples:

L105 – Sample preparation for scanning electron microscope analysis and working conditions are missing.

More details were added (Lines 109-113 in red fonts).

L109 – The ratio of broth culture/soil combination is missing. Authors should describe more in detail this.

Twenty ml of isolate culture/0.5kg dried soil (1/25; v/w) was used (the details were added in its proper places (lines 123 and 124 in red fonts).

L110 – How long after inoculation were soil used for sowing?

The inculcated soil was kept in dark at 30°C for 1 day (These details were added in red font; line 125).

L111 – seed details are missing (e.g. variety, cultivar, company). Moreover, how was the seed surface sterilization carried out?

All required details were added as requested by the reviewer (Lines 126 and 128).

L128 – Which were the appropriate media?

Required details about the appropriate media were added in materials and method section (Lines 147 and 148 in red font).

L134-141 - Metabolite profiling were these analyses performed on broth cultures? I suggest to clearly present two separated sections: one on isolates screening and another on plant extracts. At the beginning of each section should be presented the extraction procedure/the way authors get the material to be investigated with the different assays.

Under the heading 2.8. metabolite profiling (line 154) we separate the screening of isolates from the screening of the plant extracts under two separate subheadings (line 155 and line 162), and all details were provided as suggested by reviewer.

L140-141 – Revise the sentence and provide more details about methods.

This section was totally restructured, so the sentence was totally changed and improved upon request of the reviewer.

L154 – Revise the sentence.

More details were added in red font (Line 204 and 205)

L155 – I suggest to add immediately after section 2.10 title the details of plant tissues extraction. Please, provide more details about the homogenization procedure.

Require details were added. Also, additional details were added about the homogenization procedure (Lines 154-156)

L179 – Which was the standard kit used for DNA isolation? Please, provide name and company details.

The name and company of the used kit were added in lines 116 and 117 (red font).

 RESULTS AND DISCUSSION

The results are presented correctly, however, the not clear materials and methods section preclude to understand some findings. In general, the discussion is too short and is only limited to the comparison of some other authors findings. Authors should ameliorate the discussion; perhaps, a separated discussion section could help authors to ameliorate it.

Thanks for your appreciated notes. More details were added in materials and methods section. The results and discussion were separated, and we provided the discussion with more details to make it deeper and more informative. 

L199-206 – Authors should discuss more in detail the implication of some of the characteristic of the isolates. Are these characteristic important for the plant growth-promoting abilities? Can these characteristics be taken into account for the utilization of these strains for agricultural applications?

Thanks for your appreciated notes. The results and discussion were separated, and we provided the discussion with more details to make it deeper and more informative. 

L211-233 – According to the materials and methods section, the total flavonoid and phenolic contents, antioxidant, antiprotozoal and antibacterial activities were performed on plant extracts. In this section are presented results of the same assays as bioactivity screening of actinomycetes isolates. Authors should clarify the methods in materials and methods section and modify this section accordingly.  

All details on material and methods were included as suggested

L250-257 – The discussion of these findings should be ameliorated.

We provided the discussion with more details to make it deeper and more informative. 

CONCLUSIONS

This section is appropriate; however, authors should ameliorate the previous sections to get the validity of the statements made.

Thanks for your respected comments. All previous sections were intensively reviewed in the light of your valuable comments which really improve the quality of our manuscript.

 MINOR ISSUES

L46 – “They can also biosynthesize…” It is not clear the subject of the sentence. Please, rephrase.

The subject was assigned, and the sentence became clearer (lines 51-52)

L52 – “biocontrol agents, and plant growth-promoting chemicals”

The sentence was rephrased (Line 57 with red font)

L91-94 – The sentence is too long, and it is not grammatically correct. Please, rephrase.

This sentence was rephrased and incorporated in the last paragraph in the introduction to highlight the objectives of this study. This paragraph was subjected to mandatory changes (Lines 92-97, the changes were assigned in red font).

Some format issues: L95, L197 – Title; L156 – italics; L140, L176 – Subscript;

All headings and subheadings were carefully revised, and their formats were unified.

L189 – substitute “cultivar” with “crop” or “vegetal species”.

Done (Line 247)

Figure 2 quality should be ameliorated.

The original image was added in page 18 in the new revised version

L301-307 – Authors start the sentence with “The positive effect of actinomycetes on nitrogen fixation by root nodules has been reported.”, however, only one study on actinomycetes has been reported.

More details with their citations were added in Lines (420-437)

orate the quality of our MS.   

Reviewer 2 Report

The manuscript “Plant growth-promoting actinobacteria: A promising strategy for improving nutritive value of legumes by inducing the production of bioactive metabolites” by AbdElgawad et al.  demonstrates that actinomycetes can promote the growth of different legume species as well as to improve the quality of theirs seeds, by enriching the quality and fertility of the soils.

The article presents numerous data that shows the positive effect of the different actinomycetes isolates, although the effect is dependent on the leguminous plant, in the soil, plant tissues and in the nutritional content, including phenolic and flavonoids content, in the seeds. Besides that, the authors morphologically and biochemically characterized all isolates obtained and subsequently phylogenetically identified the 4 isolates selected for assays in plants. the article is well written and well read. Overall, this manuscript contributes to the better understanding of the interactions between legumes and actinobacteria. The methods performed are solid and data interpretation is appropriate. I however have several comments on the manuscript for the authors. These are listed below. 

Main points:

  • The title does not illustrate the work done, needs to be more elucidative about the main findings of the work
  • The manuscript should have separate results and discussion sections. I realize that because it has many results it seems easier to present the results and discussion together, but how it is, the discussion is superficial and but I think that a more in-depth discussion of the results as a whole and not of the results separately would significantly improve the article.

Minor points

-Indicate the objectives or question in the abstract and rewrite the sentence “this study linked actinomycetes with quality and productivity increments of legume crops by increasing nitrogen availability and metabolism, improving microbial population in soil and inducing bioactive primary and secondary metabolites and mineral accumulation”. Caution, the data does not indicate everything that is in the statement.

- Use Keywords that are not in the title

- L115: indicate the duration of the experiment for each legume plant

- L117: indicate which was the solvent for soil

- Figure 2- the image quality of figure two should be improved. it is very difficult to see the words

- Figure 3- should indicate the number of nucleotides of the sequence used to construct the tree and include an outgroup in the tree

- Tables- it may be easier to analyze or interpret if the results are grouped by legumes and not by strains

Author Response

Response to Reviewer#2

The manuscript “Plant growth-promoting actinobacteria: A promising strategy for improving nutritive value of legumes by inducing the production of bioactive metabolites” by AbdElgawad et al.  demonstrates that actinomycetes can promote the growth of different legume species as well as to improve the quality of theirs seeds, by enriching the quality and fertility of the soils. The article presents numerous data that shows the positive effect of the different actinomycetes isolates, although the effect is dependent on the leguminous plant, in the soil, plant tissues and in the nutritional content, including phenolic and flavonoids content, in the seeds. Besides that, the authors morphologically and biochemically characterized all isolates obtained and subsequently phylogenetically identified the 4 isolates selected for assays in plants. the article is well written and well read. Overall, this manuscript contributes to the better understanding of the interactions between legumes and actinobacteria. The methods performed are solid and data interpretation is appropriate. I however have several comments on the manuscript for the authors. These are listed below. 

 Main points:

Point 1: The title does not illustrate the work done, needs to be more elucidative about the main findings of the work

Response 1: The title was changed to elucidate the main findings of the study.

Point 2: The manuscript should have separate results and discussion sections. I realize that because it has many results it seems easier to present the results and discussion together, but how it is, the discussion is superficial and but I think that a more in-depth discussion of the results as a whole and not of the results separately would significantly improve the article.

Response 2: Thanks for your appreciated notes. The results and discussion were separated, and we provided the discussion with more details to make it deeper and more informative. 

Minor points

Point 3: Indicate the objectives or question in the abstract and rewrite the sentence “this study linked actinomycetes with quality and productivity increments of legume crops by increasing nitrogen availability and metabolism, improving microbial population in soil and inducing bioactive primary and secondary metabolites and mineral accumulation”. Caution, the data does not indicate everything that is in the statement.

Response 3: The sentence was rephrased (Lines 38-41)

Point 4: Use Keywords that are not in the title

Response 4: As we changed the title, the key words became a little bit different

Point 5: L115: indicate the duration of the experiment for each legume plant

Response 5: details were added (Line 135)

Point 6: L117: indicate which was the solvent for soil

Response 6: Details concerned soil analysis were added (Lines 143-148)

Point 7: Figure 2- the image quality of figure two should be improved. it is very difficult to see the words

Response 7: The original image was added in page 18 in the new revised version

Point 8: Figure 3- should indicate the number of nucleotides of the sequence used to construct the tree and include an outgroup in the tree

Response 8: Thanks for your suggestion. The number of nucleotides of the sequences used to construct the tree for the four selected isolated was added in the tree, while the tree is unrooted, and this was also mentioned in the figure legend (Page 19; lines 764-768)

Point 9: Tables- it may be easier to analyze or interpret if the results are grouped by legumes and not by strains

Response 9: Thanks for valuable comment, Done

Reviewer 3 Report

Line 48. High GC content? Out of the argument and not connected to the rest of introduction

In introduction the authors move from Actinomycetes to Streptomicetes as subject without connection. The literature review should be useful to come inside the topic of the manuscript, not for list literature without logic. Same for the genera Actinomadura, Micromonospora, 67 Streptosporangium, and Nocardia.(line 67)

It is not clear why the study is so important in the last sentences of introduction.

Line 97 even if the authors give the reference, please add some details on the soil dilution method.

Ref of the Glycerol-Yeast Extract Agar

Line 108 Twenty = 20

Line 117 EC= ?

Line 122 Content of mineral nutrients and 122 organic matter (OM) were determined, how?

Line 197 typo headline

I tried to review the manuscript in details but the corrections are too many.

I suggest to the authors a deep revision from the English language point of view and the scientific form of the manuscript. The paper must be improved for the publication. In the present form is not acceptable.

Author Response

Response to Reviewer#3

Point 1: Line 48. High G+C content? Out of the argument and not connected to the rest of introduction

Response 1: It is an introductory sentence about actinomycetes. GC-content means guanine-cytosine content which is the percentage of nitrogenous bases in a DNA or RNA molecule that are either guanine (G) or cytosine (C). The GC-content indicates the proportion of G and C bases out of an implied four total bases. The abbreviations were replaced with their full names for clarification (Line 55 in red font)

Point 2: In introduction the authors move from Actinomycetes to Streptomicetes as subject without connection. The literature review should be useful to come inside the topic of the manuscript, not for list literature without logic. Same for the genera Actinomadura, Micromonospora, Streptosporangium, and Nocardia (line 67).

Response 2: Some details were added to clarify that Streptomicetes, Actinomadura, Streptosporangium, Micromonospora and Nocardia are different genera of actinomycetes and have ameliorative actions upon plant growth (lines 70-74 in red font). Also, essential modifications were performed on the whole MS upon the request of other reviewers therefore references were subjected to crucial amendments.

Point 3: It is not clear why the study is so important in the last sentences of introduction.

Response 3: Thanks for your valuable comments. Essential modifications were made on the last paragraph of introduction to highlight the objectives of this study (Lines 94 – 99 in changes were assigned in red font)

Point 4: Line 97 even if the authors give the reference, please add some details on the soil dilution method.

Response 4: More details were added to clarify this issue (lines 103-106 with red fonts)

Point 5: Ref of the Glycerol-Yeast Extract Agar

Response 5: A reference was added

Point 6: Line 108 Twenty = 20

Response 6: Twenty ml of the actinomycete culture was used to inoculate the soil

Point 7: Line 117 EC=?

Response 7: EC= Electrical Conductivity, it was added in line 141

Point 8: Line 122 Content of mineral nutrients and organic matter (OM) were determined, how?

Response 8: Required details were added (lines 142-148)

Point 9: Line 197 typo headline

Response 9: Unfortunately, this comment is not clear for me.

Point 10: I tried to review the manuscript in detail, but the corrections are too many. I suggest to the authors a deep revision from the English language point of view and the scientific form of the manuscript. The paper must be improved for the publication. In the present form is not acceptable.

Response 10: Careful revision with Intensive modifications were applied to the whole MS especially M&M, results and discussion sections. The corrections made more clarity to our methodologies and add more highlight to our findings. We appreciate your effort to ameliorate the quality of our MS.

Round 2

Reviewer 1 Report

Authors correctly adressed all previous issues. There still are some other minor changes required:

  • text formatting
  • L224 - revise "..and the superannuants were.."
  • L260 - newline for paragraph 3.1
  • L57 - remove space between "/" and "or"
  • L418 - provide "first author name et al."

Author Response

"Examples of grammar problems were listed below and there are still many more
should be checked by the authors:
Point 1: Line 29: to genus Streptomyces - to the genus Streptomyces.

Response 1: Done  
Line 34: nitrogen containing amnio acids - nitrogen-containing amino acids

Response 2: Done 
Point 3: Line 40: our data reinforces - our data reinforce

Response 3: Done 
Point 4: Line 367: expect isolate 15 - except for isolate 15

Response 4: 
Point 5: Line 477: marco - macro

Response 5: Done